# Anatomical Variations of *Modiolus* in Relation with Vestibular and Cranial Morphology on CT Scans

Caroline Guigou [1,2,*], Raabid Hussain [2], Alain Lalande [2,3] and Alexis Bozorg Grayeli [1,2]

1 Department of Otolaryngology-Head and Neck Surgery, Dijon University Hospital, 21000 Dijon, France
2 ImViA Laboratory, Burgundy-Franche-Comte University, 21000 Dijon, France
3 Department of Medical Imaging, Dijon University Hospital, 21000 Dijon, France
* Correspondence: caroline.guigou@chu-dijon.fr; Tel.: +33-(61)-5718531

**Abstract:** Background: Fundamental knowledge of the anatomy and physiology of the inner ear is necessary to understand otologic diseases and therapeutic strategies. Aim: Evaluate the inter- and intraindividual variability of the modiolar position in relation to vestibular landmarks and cranial morphology on computed tomography scans (CT scan). Methods: Thirty CT scans of normal temporal bones (25 adults, 5 children) were analyzed after multiplanar reconstruction (MPR). The measurements for each ear included the angle of each semicircular canal (SCC) made by a line passing through the chosen plane and a line passing between the apex and the *ampulla* of the SCC studied and the angle of the *modiolus* in the transverse and sagittal planes. Results: Intraindividual asymmetries with a moderate to good right/left correlation were observed for the lateral SCC in the transverse plane, posterior SCC in the frontal plane, and the superior SCC in the sagittal plane and for the *modiolus* in the transverse plane. Conclusions: An anatomical variability in the cochlea, independent of other surrounding anatomical elements, seems to exist, but the SCCs seem to remain symmetrical. Significance: The orientation of the *modiolus* is an important knowledge to acquire during presurgical planning prior to transmodiolar auditory nerve implantation.

**Keywords:** presurgical planning; temporal bone imaging; semicircular canals; *modiolus*

## 1. Introduction

Fundamental knowledge of the anatomy and physiology of the inner ear is necessary in order to understand various otologic diseases and therapeutic strategies. Anatomical studies have been performed to describe the structure and dimensions of the inner ear [1–4]. Most of these studies agree that the three semicircular canals (SCCs) are perpendicular to each other with the persistent hypothesis of an average angulation of 100° between the anterior and lateral SCCs [3,4]. Lengths of the three SCCs differ, with an average of 16.72 mm for the posterior SCC, 15.05 mm for the superior SCC, and 12.57 mm for the lateral SCC [1]. Their diameter has been measured at 0.14 mm and they contain 0.2 mm$^3$ of endolymph. To the best of our knowledge, the spatial orientation of the three SCCs, their inter- and intraindividual variabilities, and their interaural correlation have been, to date, very imperfectly described. Identifying the precise position of the SCCs in the three spatial planes is important to the extent that this position could have consequences on a subject's posture and because certain variabilities might explain or promote some instabilities or balance disorders. This is because the SCCs encode for rotational movements of the head and provide sensory afferences to the vestibulo-ocular reflex (VOR), vestibulo-collic reflex, vestibulo-spinal system, vestibulo-reticular system, cerebellum, and cortex [5,6]. SCC stimulation is maximal when the head rotates in its plane and is minimal when the rotation is in a perpendicular plane [3]. An improved understanding of the position of the SCCs in space, based on a scannographic study, could make it possible to adapt vestibular functional explorations, in particular, head movements during VOR studies using the video head

impulse test (VHIT). Numerous studies have focused on the anatomy of the cochlea [4,7–10]. These studies have shown a great variability of the anatomy but with concordant results. The methods of study were varied: either by indirect methods by multiple sections of the cochlea or by scannographic or μCT studies [4,7–10]. The cochlear duct has a length of between 3.2–4.2 cm with high interindividual variability (relative standard deviation (RSD) > 1000% for the length of the basal turn and RSD > 800% for the total visible length of the cochlear duct) [4,7,8]. The cochleae studied had $2.6 \pm 0.17$ turns of spiral (extreme: 2.39; 2.84) which corresponds to $949 \pm 2.84$ degrees [9]. A total of 74% of cochleae have more than 2.5 turns of spiral and 13% have 2.5 turns of spiral [8]. The cochlea basal turn accounts for about 53% of the total length of the cochlear duct. The cochlear duct has a length between 9.1 to 22.6 mm [4,7,9]. The diameter of the cochlear canal tends to decrease progressively but not linearly from the base to the cochlear apex (the 180° measurement is between 1.14 and 1.72 mm; the 380° measurement is between 1.09 and 0.71 mm and the 540° measurement is between 0.9 and 0.58 mm) [7,9]. The length of the modiolus has been measured at $5.3 \pm 0.54$ mm with a surface area of $4 \pm 0.4$ mm$^2$ [11]. The weight of the cochlea was $4.4 \pm 0.3$ grams [9]. In contrast, the orientation of the cochlea, and thus of the modiolus, in space has received even less attention in the literature than the orientation of the SCCs.

Knowledge of the orientation of the *modiolus* is nonetheless essential during presurgical planning for transmodiolar hearing nerve implantation. Thanks to the advances made possible by augmented reality [12], research work is underway on minimally invasive surgery for transmodiolar implantation. Such transmodiolar implantations could result in a decrease in energy consumption, improved specificity of neuronal stimulation, less interference between electrodes, less loss of stimulation, and greater frequency stimulation, particularly in terms of the apex frequencies that cannot always be stimulated with conventionally inserted electrode arrays [13]. This type of implantation could also represent an elegant solution in the case of ossified or malformed cochleae. Work toward such implantation began a few years ago but was abandoned because the approach was too invasive [13]. During an anatomical study of a computed tomography scan (CT scan) on normal adult and child temporal bones (n = 122), it was shown that access to the *helicotrema* was possible in all cases. The distance between the cochleostomy and the posterior wall of the intrapetrous carotid artery was measured at $4.3 \pm 1.35$ mm in adults (n = 69) and $3.8 \pm 1.33$ mm in children (n = 53); the distance between the cochleostomy and the posterior limit of the temporomandibular joint capsule was measured at $7.8 \pm 0.2$ mm in adults (n = 69) and $5.9 \pm 0.14$ mm in children (n = 53) [14]. In the same study, computer-assisted transmodiolar implantation was successfully carried out on adult cadaveric temporal bones using a bent piezoelectric motor and after radical mastoidectomy [14]. In a more recent study, it was demonstrated that transmodiolar implantation was possible on eight adult and pediatric phantom temporal bones thanks to augmented reality, which provided indications on the entry point of the modiolus as well as its axis [12]. One of the limitations of this work was the difficult access to the *helicotrema* due to an anterior space limiting the mobilization of the burr and the Rosen needle. Precise measurement of the modiolus axis during presurgical planning would allow for better selection of patient eligibility for this new type of cochlear implantation.

Working from the hypothesis that there may be positional variability of the inner ear and the axis of the temporal bones relative to the three planes of space with consequences on clinical practice, it appeared interesting to carry out a descriptive human CT scan anatomical study. The aim of the present work was to evaluate the positional variability of the 3 semicircular canals and the modiolus by means of a scannographic study on normal adult and pediatric human temporal bones.

## 2. Materials and Methods

A total of 30 patients (25 adults and 5 children) undergoing high-resolution temporal bone CT scans were included in this prospective study. The 60 temporal bones (30 right and 30 left) were analyzed by 1 observer who is an experienced otologist.

High-resolution computer tomography scans (CT scans) were obtained by helical acquisition with a 0.6-mm slice thickness overlapping every 0.3 mm (Light Speed, 64 detector rows; General Electric Medical Systems, Buc, France). Digital imaging and communications in medicine (DICOM) data were imported into an Osirix® viewer (32-bit v.5.6, Pixmeo, Geneva, Switzerland).

The measurements were performed on CT scan views after orthogonal multiplanar reconstruction (MPR). On the transverse view, the sagittal reference plane (SRP) was set to pass through the vomer and the external occipital crest. On the frontal view, the transverse reference plane (TRP) was the plane passing through the dorsal edges of the right and left internal auditory meatus. The frontal reference plane (FRP) was orthogonal to both reference planes.

The angle between the modiolar axis and FRP was measured in the transverse views (Figure 1A), and the angle between the modiolar axis and the TRP was measured in the sagittal view (Figure 1B). In order to determine the axis of the modiolus, we scrolled the CT scans in the three planes of space defined for this study in order to localize the cochlear nerve in the cochlea as best as possible. The axis of the modiolus was defined by a line passing from the cochlear apex to the cochlear fossa in the internal auditory meatus. Once the latter was found, the axis of the modiolus was then traced in the desired planes.

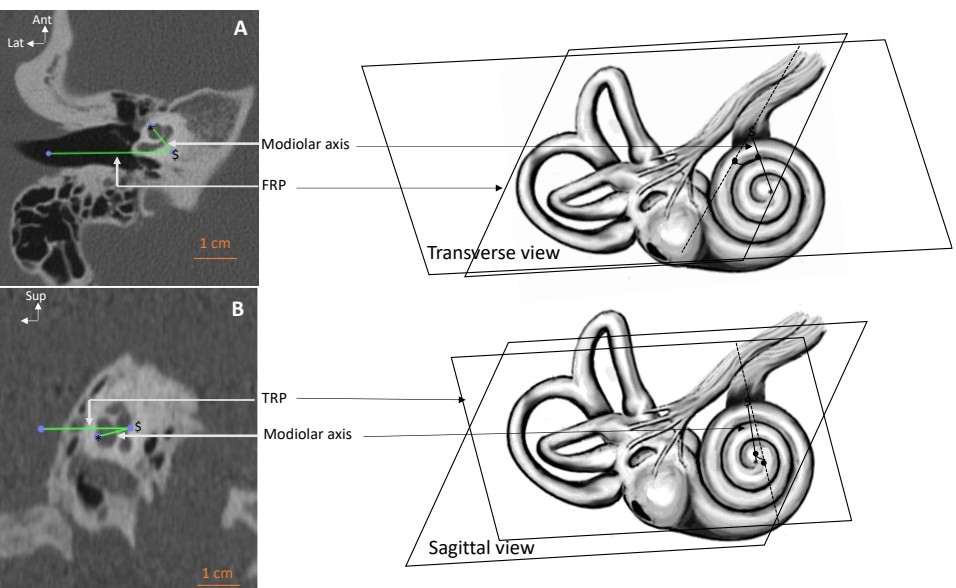

**Figure 1.** Measurement of the modiolar angle in the transverse (**A**) and the sagittal views (**B**). The axis of the *modiolus* was defined by a line passing from the cochlear apex (*) to the cochlear fossa in internal auditory meatus ($). FRP: frontal reference plane. TRP: transverse reference plane.

For each SCC, a line passing through the most distant point of the canal from the vestibule and the center of its ampulla in the selected view was drawn and its angle with the reference orthogonal planes was measured. Because of the risk of measurement bias, all these measurements were voluntarily performed by a single observer who always located the line between the ampulla and the most distal point of the SSC in the same line. Lateral SCC measurements were conducted in the frontal and the transverse planes (Figure 2).

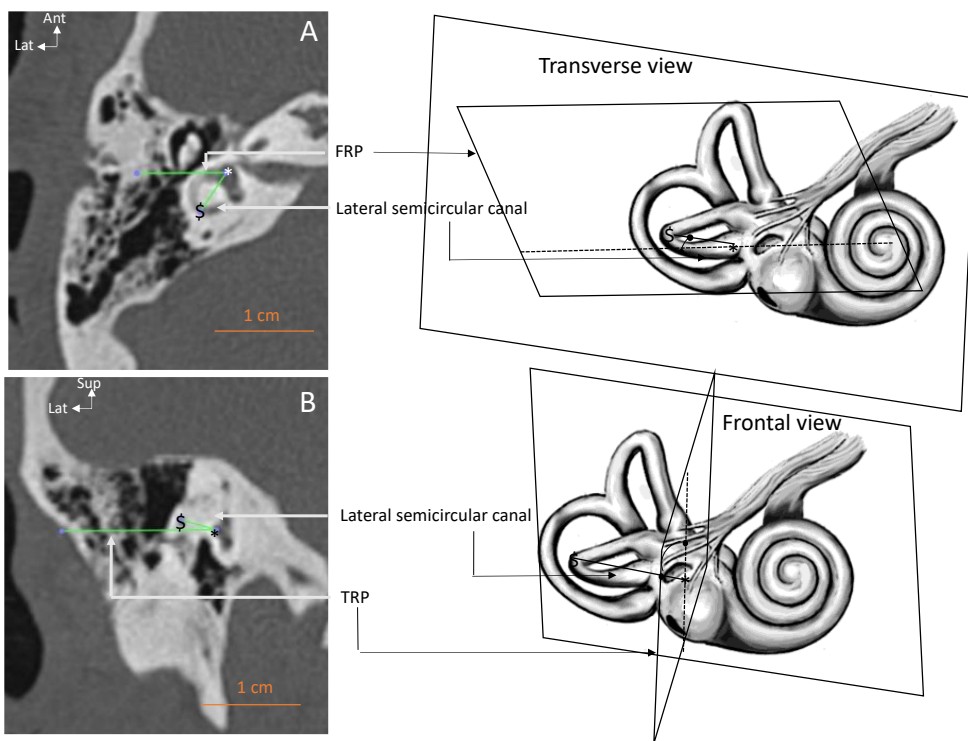

**Figure 2.** Angle measurements for the lateral semicircular canal (SCC) in the transverse (**A**) and frontal (**B**) views. Angles are measured between a line passing between the canal apex and the center of its ampulla (*) and its projection on each plane ($). FRP: frontal reference plane. TRP: transverse reference plane.

In the transverse view, the angle between the lateral SCC and FRP was measured. In the frontal view, the angle between the lateral SCC and TRP was measured. The angles of anterior (Figure 3) and posterior SCC (Figure 4) were measured in sagittal and the frontal planes with the TRP. Angles for SCCs and modiolus could not be measured in all three planes because the measurements were close to the precision limits provided by the CT scan as the sections were thicker during the MPR reconstruction.

The rearward open interpetrous angle formed by two lines passing through the center of the horizontal portion of the internal carotid artery (Figure 5) and the intervestibular distance (defined as the distance between the two centers of the anterior osseous ampullae of the anterior SCCs) were also recorded.

Values were expressed as means ± SD [min-max]. To evaluate the variability of the parameters, the relative standard deviation (RDS, %) was estimated. The RSD is defined as the absolute value of the coefficient of variation around the mean. Intraindividual differences were compared within each individual (right versus left). The comparison between individuals allowed the study of interindividual variations. Paired and unpaired Student *t*-tests were carried out to evaluate intra- or interindividual differences. A *p*-value < 0.05 was considered significant. Pearson's test (r) followed by Fisher's test was employed for correlation analysis. For R-values in the range of 0.51–1.0, correlations were considered strong, r in the range of 0.31–0.5 indicated moderate correlation, and values <0.1 were considered as low. Statistical tests were performed on Prism software (v.6, GraphPad, San Diego, CA, USA).

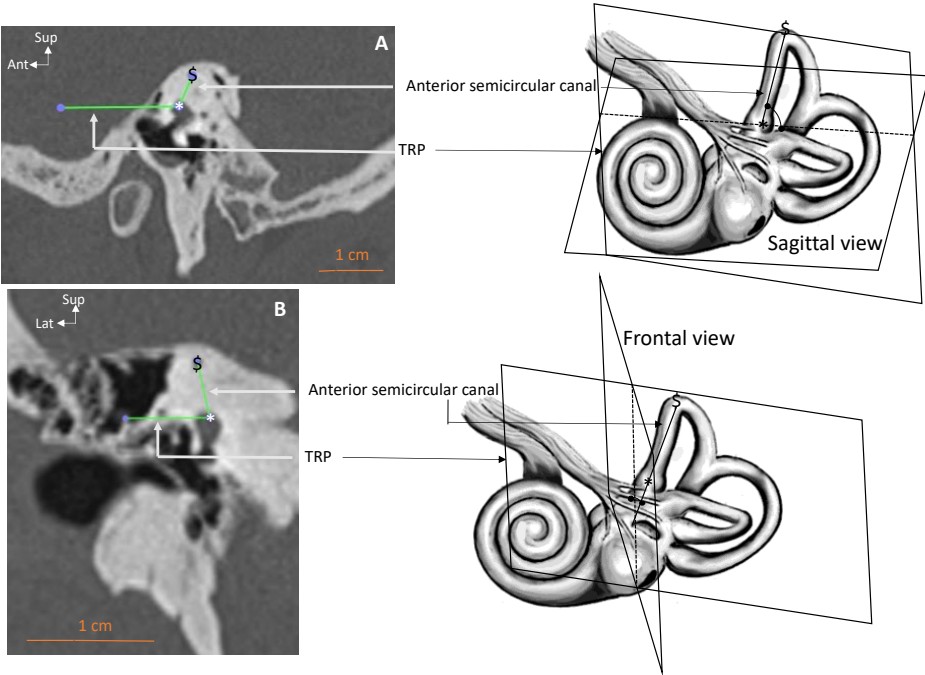

**Figure 3.** Angle measurements for the anterior semicircular canal (SCC) in the sagittal (**A**) and the frontal views (**B**). Angles are measured between a line passing between the canal apex and the center of its ampulla (*) and its projection on each plane ($). TRP: transverse reference plane.

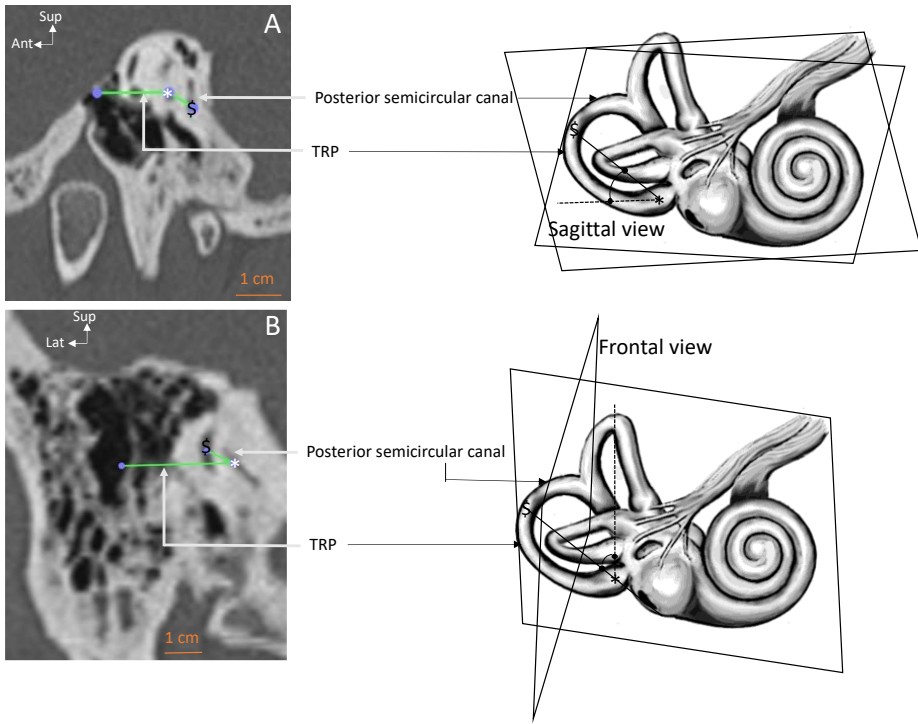

**Figure 4.** Angle measurements for the posterior semicircular canal (SCC) in the transverse (**A**) and frontal (**B**) views. Angles are measured between a line passing between the canal apex and the center of its ampulla (*) and its projection on each plane ($). TRP: transverse reference plane.

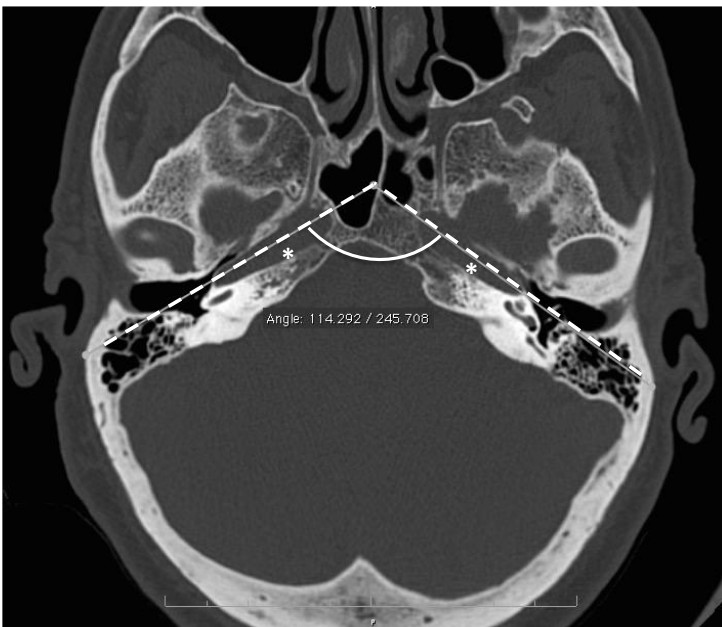

**Figure 5.** Measurement of the interpetrous angle in the transverse plane. The measured angle opens towards the back (formed by 2 lines passing through the center of the internal carotid artery with the vomer axis as the apex). *: internal carotid artery.

## 3. Results

The population comprised 14 women and 16 men. The mean age of patients included in the present study was $45 \pm 21.2$ years [8–76] ($46 \pm 21.2$ years for women and $42 \pm 22.5$ years for men, $p = 0.8$). The RSD indicating the relative dispersion of the values was generally low, except for lateral and posterior SCC angles in the frontal plane and for the modiolar angle in the sagittal plane (Table 1 and Figure 6).

**Table 1.** Radiological measurements of inner ear structures on high-resolution CT scans: modiolar axis, semicircular canals, intervestibular distance, and interpetrous angle measurements on three orthogonal conventional multiplanar plan reconstructions. Values are expressed as means $\pm$ standard deviation [min-max]. Distances are expressed in centimeters and angles in degrees. SCC: semicircular canals. RSD: relative standard deviation, expressed as a percentage, *: $p < 0.05$ (paired *t*-test, right versus left ears, n = 60).

| Parameter–View | All Ears | Right Ears | Left Ears | RSD |
|---|---|---|---|---|
| Intervestibular distance (cm) | $7.8 \pm 0.38$ [6.9–8.7] | - | - | 5 |
| Interpetrous angle (deg.) | $107 \pm 3.7$ [94–123] | - | - | 3 |
| *Modiolus*–transverse (deg.) | $52.5 \pm 9.74$ [30.8–70.3] | $54.7 \pm 6.75$ * | $50.3 \pm 5.56$ | 19 |
| *Modiolus*–sagittal (deg.) | $17.1 \pm 8.09$ [1–50.5] | $17.4 \pm 5.77$ | $16.8 \pm 5.27$ | 47 |
| Lateral SCC–transverse (deg.) | $62.3 \pm 7.31$ [43.1–75.6] | $63.9 \pm 7.39$ * | $60.8 \pm 9.15$ | 11.7 |
| Lateral SCC–frontal (deg.) | $6.8 \pm 3.86$ [1–17] | $7.1 \pm 5.21$ | $6.5 \pm 2.01$ | 57 |
| Posterior SCC–sagittal (deg.) | $142.6 \pm 11.3$ [118.2–169.4] | $142.2 \pm 16.79$ | $142.9 \pm 10.25$ | 8 |
| Posterior SCC–frontal (deg.) | $16.9 \pm 7.34$ [5–43.5] | $18.4 \pm 14.93$ * | $15.3 \pm 8.64$ | 43 |
| Superior SCC–frontal (deg.) | $75.3 \pm 9.81$ [53.7–107.2] | $76.7 \pm 9.61$ | $73.9 \pm 6.79$ | 13 |
| Superior SCC–sagittal (deg.) | $111.4 \pm 12.53$ [73.7–135.1] | $114.5 \pm 7.74$ * | $108.4 \pm 17.88$ | 11 |

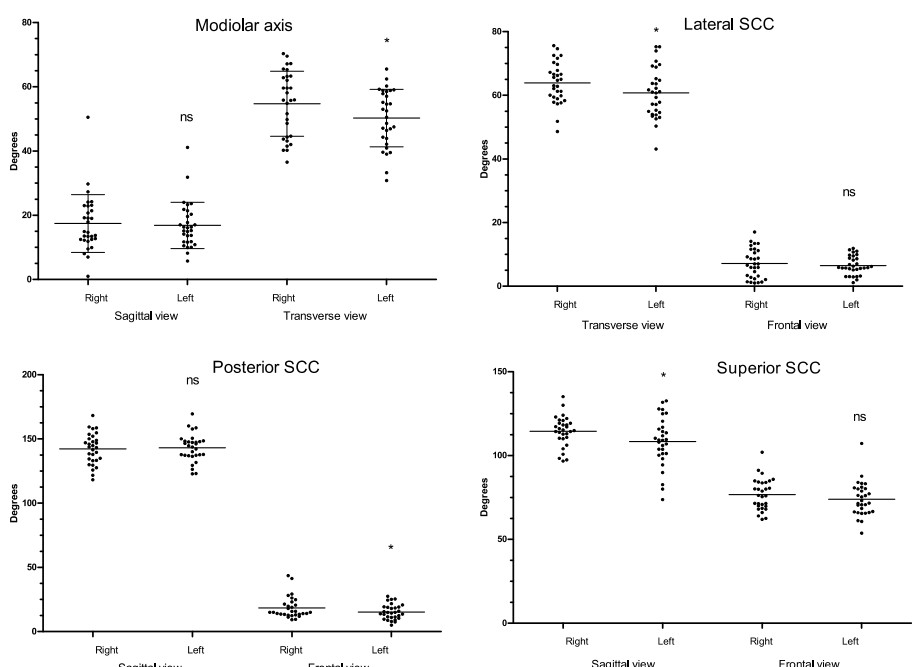

**Figure 6.** Dispersion of the values of the modiolus and the three right and left semicircular canals. SCC: semicircular canals. *: *p* < 0.05 (paired *t*-test, right versus left ears, n = 60), ns = not significant.

Age did not seem to influence the variables as judged by Pearson's correlation index. Men had a larger intervestibular distance than women (107.8 ± 1.94 mm versus 106.0 ± 2.31, respectively, *p* < 0.05 unpaired *t*-test). The sex did not seem to influence other measurements. The modiolar axis appeared to be asymmetric in the transverse plane with a higher angle on the right side in comparison to the left (*p* < 0.05, paired *t*-test, n = 60, Table 1). Asymmetries were also observed for the lateral SCC in the transverse plane, posterior SCC in the frontal plane, and the superior SCC in the sagittal plane with larger angles on the right than on the left (*p* < 0.05, paired *t*-test, n = 60) (Table 1, Figure 6). Despite side differences in average values, a moderate to good right–left correlation was observed for the modiolar axis and for the SCCs except for the superior SCC in the frontal plane (Table 2).

**Table 2.** Right–left correlations for modular axis and semicircular canal (SCC) angles. Correlations were judged as strong for Pearson's r in the 0.51–1.0 range, as moderate for 0.31–0.5, and as weak for values between 0.1 and 0.3 (n = 60).

| Parameter–Plane | Pearson's r | 95% Confidence Interval | *p*-Value (Fisher's r to z) |
|---|---|---|---|
| *Modiolus*–Sagittal | 0.686 | 0.432–0.839 | <0.0001 |
| *Modiolus*–Transverse | 0.580 | 0.277–0.778 | 0.0006 |
| Lateral SCC–Frontal | 0.344 | −0.190–0.627 | 0.0625 |
| Lateral SCC–Transverse | 0.602 | 0.309–0.791 | 0.0003 |
| Posterior SCC–Frontal | 0.622 | 0.377–0.803 | 0.0002 |
| Posterior SCC–Sagittal | 0.694 | 0.446–0.844 | *p* < 0.0001 |
| Superior SCC–Frontal | 0.239 | −0.133–0.552 | 0.2 |
| Superior SCC–Sagittal | 0.536 | 0.218–0.751 | 0.0019 |

Interestingly, the modiolar axis in the sagittal plane appeared to be moderately correlated to the superior SCC angle in the frontal plane (Pearson's r = 0.356, 95% confidence interval: 0.112–0.559, *p* < 0.01 Fisher's test). There was no other correlation between the modiolar axis in the sagittal plane and any other parameter. Similarly, the modiolar axis in the transverse plane was not correlated to any other parameter.

## 4. Discussion

In this study, we observed a relatively high variability of the modiolar orientation in the sagittal plane, whereas its orientation in the transverse plane is fairly constant. Although left and right measurements regarding SCC and *modiolus* had a good to fair correlation, we noted an asymmetry regarding the SCC and the *modiolus* angles in the transverse plane. The modiolar axis in the sagittal plane seemed to be correlated to the superior SCC angle measured in the frontal plane but not to interpetrous angle or intervestibular distance suggesting that its geometry is more dependent on other inner ear elements than on the petrous bone position. No significant to moderate correlations between the modiolar measurements and those of SCC or petrous bone axis also suggest that the *modiolus* axis develops in its position independently from surrounding anatomical elements. This notion is interesting, and to our knowledge, it has not been described in the literature before. The results for some angles indeed show great variability. We redid the measurements with extreme values to check them and found that they were correct. We decided to keep them for more transparency and to show the variability of the position of some SCCs in space.

As for the *modiolus*, the position of the SCCs in space seems to correlate well between sides with a small interaural difference. This correlation has already been noted, with an interaural difference of approximately 19° for the lateral SCCs and 23° for the vertical SCCs [3]. It is important to take this factor into account in clinical practice, especially during the study of VOR and the head impulse test. The orientation of some SCCs does not seem to depend on the cranial morphology (no correlation between the interpetrous angle and the SCCs in the three planes). These results are interesting from the anatomical standpoint since they are in contrast with the influence of the petrous bone angle on the mastoid bone development and the sigmoid sinus position [15].

The lack of correlation between age and intervestibular distance may be due to the small population of children in this study. Ethnic factors may also participate in the variability of this angle and the size of the posterior fossa [16]. Measured by magnetic resonance imaging (MRI), the distance from the craniocaudal axis of the internal auditory canal through the transition from the transverse to the sigmoid sinus was reported in 60 Asian, 57 African American, and 70 Caucasian individuals. The junction between the transverse sinus and the sigmoid sinus appeared to be shorter in Caucasians (5.5 mm) than in Asians (7.3 mm) and African Americans (10.6 mm). This finding could have an impact on neurosurgical approaches, particularly retrosigmoid approaches. Ethnicity may be a predictor of posterior fossa volume as well as the body mass index (BMI) and sex [17]. The volume of the posterior fossa seems to be greater in white subjects (1.77 $mm^3$ more than in black subjects), in men (5.74 $mm^3$ more than in women), and in subjects with high BMI (0.39 $mm^3$ per 10 kg/$m^2$) [17]. The interpetrous angle could also be related to the risk of chronic otitis media. Indeed, the interpetrous angle seems to be larger in patients with chronic otitis media compared to healthy subjects, a fact which could influence the angle of the Eustachian tube isthmus and its patency (average interpetrous angle in healthy subjects: 106.7° (n = 41), versus 114.4° in patients with central tympanic perforations (n = 45), and 120.5° in patients with atrial or posterior superior tympanic perforations (n = 66)) [18]. In this study, the interpetrous angle was measured between two lines passing along the back edge of the two petrous blocks. On the other hand, no impact of sex was found on the interpetrous angle [18].

In a previous study, we evaluated the feasibility of auditory nerve implantation via the middle ear and *helicotrema* [14]. The orientation of the *modiolus* is an important parameter to ascertain during presurgical planning before a transmodiolar implantation. We showed that various anatomical measurements on temporal bone CT scans were compatible with auditory nerve implantation by the transmodiolar route [14]. However, its variability and the absence of reliable anatomical landmarks warrant the use of computer-assisted navigation by augmented reality [12]. The preoperative study of the *modiolus* axis would aid in the selection of patients eligible for this type of implantation.

We did not measure the angle of the SCCs with respect to each other, as the objective here was not to conduct an intravestibular anatomical study, but rather to study the position of different anatomical elements of the otic capsule in relation to the external environment and cranial morphology. In previous studies, no side difference was shown in the orientation of the SCCs, which is consistent with our results ($p > 0.57$ on multiway ANOVA of orientation vector coefficients, n = 22) [2,3]. The position of each SCC was studied in space with respect to Reid's planes [2,3]. Reid's horizontal plane (Z Reid) corresponds to a plane passing through the center of the two external auditory canals and the inferior margin of the orbits. Y Reid (the sagittal plane) passes through the mid-sagittal suture and is perpendicular to the interaural axis. X Reid (the coronal plane) contains the interaural axis and is perpendicular to Z Reid [2,3]. The lateral SCC is the canal whose position with respect to the environment has been the most fully studied: it seems to form an angle of about 15.9° with respect to the Frankfurt plane (which corresponds to the plane passing through the floor of the orbit and above the external auditory canal) [19] and an angle ranging from 15° to 30° with respect to the horizontal plane [3]. To compare the results of these various studies, the authors calculated that there is a probable variation of approximately 4.3° degrees between Reid's horizontal plane and the Frankfurt plane [3]. Our results cannot be directly compared to these studies since we did not choose the same planes in space, nor did we choose the same method of measuring angles. Another limitation of our work is that the measurements were made on the bony and not on the membranous vestibule. It is widely accepted that membranous SCCs adhere to the lateral walls of bone canals at the point of the maximum bone radius of the canal with possible deviations and deviations from the predominant central axis in the case of the lateral SCC [1,20]. Based on biophysical measurements, the maximum angle between the membranous and bony canals is estimated to be 5.4° [20]. This variation in angulation remains acceptable in the context of clinical applications. Nevertheless, with the evolution of technology, measurements can increasingly be performed directly on the membranous labyrinth and no longer by extrapolation from the bone labyrinth thanks to μCTs and to manual or semiautomatic segmentation [1].

The software that we used is commercially available and accessible to otologists and serves for the preplanning of complex surgeries on inner ear structures. We have already used this software in previous work on the anatomical study of the cochlear duct [8].

Finally, our angle measurements are prone to errors related to the section thickness of routine CT scans. This phenomenon could have added random noise to the data. However, the results were consistent with previously published anatomical data (intervestibular distance for example) [17] and fair to good left–right correlations for the majority of the parameters suggested that this random noise did not disturb the observations.

## 5. Conclusions

In conclusion, this study highlighted the lack of impact of skull anatomy in the positioning of the SCCs and the *modiolus,* and a moderate to good right–left correlation of SCCs and the *modiolus* positions. Furthermore, it suggests that the *modiolus* angle is independent or has a low dependency on the disposition of other surrounding anatomical elements. These findings are especially interesting in the context of developing minimally invasive techniques for transmodiolar auditory nerve implantation.

**Author Contributions:** Conceptualization, C.G. and A.B.G.; methodology, C.G. and A.B.G.; valida-tion, C.G., R.H., A.L. and A.B.G.; investigation, C.G. and A.B.G.; resources, C.G. and A.B.G.; data curation, C.G.; writing—original draft preparation, C.G., R.H., A.B.G. and A.L.; writing—review and editing, C.G., A.B.G., A.L. and R.H.; visualization, A.B.G.; supervision, A.B.G. and A.L. All authors have read and agreed to the published version of the manuscript.

**Funding:** This research received no external funding.

**Institutional Review Board Statement:** Not applicable.

**Informed Consent Statement:** Informed consent was obtained from all subjects involved in this study.

**Data Availability Statement:** Data is unavailable due to privacy or ethical restrictions, a statement is still required. If you want to have access to data, please contact the corresponding author: caroline.guigou@chu-dijon.fr.

**Conflicts of Interest:** The authors declare no conflict of interest.

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
