# Peer review of "Anatomical Variations of Modiolus in Relation with Vestibular and Cranial Morphology on CT Scans"

_2813-0545, doi:10.3390/anatomia2010009_

Round 1

Reviewer 1 Report

The manuscript certainly complements the still incomplete knowledge about the morphology of the inner ear and its variability. It could provide relevant reference data. Unfortunately, too small a group size does not allow for a reliable analysis of differences depending on sex, age, and other factors.

The anatomical description of individual structures and organs published, among others, in this journal Anatomy, often concerns animal models, especially in the case of the inner ear. Therefore, I suggest, if not in the title, then at least in the Introduction section, include a clear indication that the research concerns human subjects.

Other minor notes below:

Line 35: mm2 is a measure of surface area, not volume. Explain or correct.

line 108:  replace "cresta" by "crista". I suggest to use English name (external occipital crest) all other anatomical names are in English.

Line 109: dorsal edges of the right and left internal..

Line 110: Do you mean meatus?

Line 113: Add "axis" after "modiolar" 

Line 114: I do not understand "we scrolled the cochlea"

Line 115: What is a "modiolar nerve"? Do you mean "cochlear nerve"?

Line 156: Anterior osseous ampullae

Author Response

Thank you very much for your comments and corrections.
We hope we have answered them correctly.

Reviewer 2 Report

The manuscript entitled "Anatomical Variations of Modiolus in Relation with Vestibular and Cranial Morphology on CT-Scans". The authors have a clear objective which was to evaluate the positional variability of the 3 semicircular canals and the modiolus by means of a scannographic study on normal adult and pediatric temporal bones, for this they used a suitable material and have obtained some very interesting conclusions in the context of developing minimally invasive for Techniques for transmodiolar auditory nerve implantation.

Author Response

Thank you very much for your comments.
We have added in the introduction and summary the fact that the scanners are human scanners.